# Impact of Intervertebral Disc Degeneration and Endplate Changes on Cefazolin Penetration into the Intervertebral Disc

**DOI:** 10.3390/medicina61111999

**Published:** 2025-11-07

**Authors:** Aleksejs Repnikovs, Kalvis Briuks, Artūrs Paulausks, Pēteris Studers, Konstantīns Logviss, Baiba Mauriņa, Dace Bandere, Jānis Kurlovičs, Sigita Kazūne

**Affiliations:** 1Department of Orthopaedics, Faculty of Medicine, Riga Stradins University, LV-1007 Riga, Latvia; peteris.studers@rsu.lv; 2Department of Spine and Joint Surgery, Hospital of Traumatology and Orthopaedics, LV-1005 Riga, Latvia; 3Laboratory of Finished Dosage Forms, Faculty of Pharmacy, Riga Stradins University, LV-1007 Riga, Latviakonstantins.logviss@rsu.lv (K.L.); 4Baltic Biomaterials Centre of Excellence, Headquarters at Riga Technical University, LV-1048 Riga, Latvia; baiba.maurina@rsu.lv (B.M.);; 5Joint Laboratory of Traumatology and Orthopaedics, Riga Stradins University, LV-1007 Riga, Latvia; 6Department of Applied Pharmacy, Faculty of Pharmacy, Riga Stradins University, LV-1007 Riga, Latvia; 7Bioinformatics Group, Riga Stradins University, LV-1007 Riga, Latvia; 8Department of Anesthesiology and Intensive Care, Riga Stradins University, LV-1007 Riga, Latvia; 9Department of Anesthesiology, Hospital of Traumatology and Orthopedics, LV-1005 Riga, Latvia

**Keywords:** cefazolin, antibiotic prophylaxis, degenerative disc disease, spine surgery

## Abstract

*Background and Objectives*: Preoperative cefazolin is the standard of care for intervertebral disc surgery as it reduces the incidence of iatrogenic spondylodiscitis. The aim of this study was to determine the impact of intervertebral disc degeneration and endplate changes on the penetration of prophylactic cefazolin into the intervertebral disc during spinal surgery. *Materials and Methods*: Adult patients undergoing single-level microdiscectomy for lumbar disc herniation received prophylaxis with 2 g of cefazolin. Venous blood and intervertebral disc samples were collected and analyzed using high-performance liquid chromatography to determine cefazolin concentrations. The severity of intervertebral disc and endplate changes was assessed on magnetic resonance images using the Pfirrmann and Modic grading systems. *Results*: Cefazolin concentrations were significantly higher in cases with Modic type II changes compared to type 0/I (14.6 ± 9.2 µg g^−1^ vs. 10.2 ± 4.5 µg g^−1^ and 9.2 ± 4.1 µg g^−1^; *p* = 0.01). 35.4% of patients with Modic type II changes had concentrations > 16 µg g^−1^, compared to 10% and 25% for patients with Modic type 0/I (*p* = 0.008). For Pfirrmann grading, 34.6% of grade V discs reached >16 µg g^−1^ versus 16.7% and 20.3% for grades III and IV (*p* = 0.26). Patient age, weight, and timing showed no significant correlations with intradisc concentrations. *Conclusions*: Ninety-four percent of disc samples exceeded the minimum inhibitory concentration for *Staphylococcus aureus* (>4 µg/g), but considerable variability in cefazolin levels was observed, with higher concentrations in discs showing Modic type II changes.

## 1. Introduction

Intervertebral disc (IVD) infections occur when bacteria spread into the disc space due to bacteremia or as a consequence of spinal surgery. After discectomy, surgical site infections (SSI) rank third among the most frequent complications, with cerebrospinal fluid leak and recurrent herniation being the two most common [1]. The European Centre for Disease Prevention and Control reported a pooled incidence rate of SSI after laminectomy and discectomy at 0.8 [0.2–2.7]% among 23,950 patients, with spondylodiscitis accounting for 54% of cases [1]. Although the incidence is low, SSI is associated with severe pain, prolonged hospitalization and can result in major disability.

The current standard of care for IVD surgery includes perioperative prophylactic antibiotics, which have been shown to reduce the risk of iatrogenic discitis [2,3]. First- or second-generation cephalosporins are recommended as first-line agents for surgical prophylaxis [4], with cefazolin widely adopted for its efficacy against common Gram-positive pathogens, favorable safety profile, and cost-effectiveness. However, the pharmacokinetics of cefazolin shows considerable interindividual variability in both detection period and tissue concentration after prophylactic dosing [5,6,7,8]. Because IVDs are avascular, antibiotics reach the disc primarily through passive diffusion across the cartilaginous endplates and into the extracellular matrix [9]. This limited vascular supply greatly restricts cefazolin delivery and slows equilibration with systemic circulation. In addition, the negatively charged extracellular matrix of the disc can further hinder diffusion of negatively charged antibiotics such as cefazolin.

The current standard of care for IVD surgery involves prophylactic antibiotics, which have been demonstrated to reduce iatrogenic discitis [2,3]. First or second generation cephalosporins are the first-line choice for perioperative prophylaxis, with cefazolin widely adopted for its efficacy against common Gram-positive pathogens, favorable safety profile, and cost-effectiveness [4]. However, the pharmacokinetics of cefazolin shows considerable interindividual variability in both detection period and tissue concentration after prophylactic dosing [5,6,7,8]. As healthy IVDs are avascular, antibiotics must be transported by diffusion from blood vessels through the endplates and then through the extracellular matrix of the disc [9]. This restricts cefazolin delivery and slows equilibration with systemic circulation. In addition, the extracellular matrix of the disc which contains proteoglycans and negatively charged glycosaminoglycans can further hinder diffusion of negatively charged antibiotics such as cefazolin. These barriers to cefazolin diffusion into IVD are affected by disc degeneration and endplate pathology. Disc degeneration leads to loss of negative charge, whereas endplate microfractures and neovascularity may modify permeability and facilitate cefazolin transport. These factors raise concerns regarding the adequacy of standard prophylactic regimens in achieving effective antibiotic concentrations within IVD during spinal surgery. While computational models and in vitro studies indicate that IVD charge and endplate permeability influence cefazolin diffusion dynamics, no clinical studies have yet examined how these factors affect antibiotic penetration in vivo [8,10].

Magnetic resonance imaging (MRI) is a useful tool for investigating IVD degeneration and structural endplate changes. Disc morphology and hydration can be graded on MRI T2 spin-echo weighted images using Pfirrmann’s grading system (grade 1–5) [11]. According to Modic et al. [12], three types of lumbar vertebral body marrow and endplate lesions can be identified on MRI [13]. From histological studies of surgical material, these lesions represent endplate fissuring, with subsequent subchondral sclerosis formation which can increase or reduce substance penetration depending on the stage of degeneration. The relationship between antibiotic penetration and MRI changes in IVDs and endplates has not been investigated, despite potential clinical implications.

We hypothesized that cefazolin IVD penetration after prophylactic preoperative administration varies according to the degree of disc degeneration and the presence and type of endplate changes. The aim of this study was to quantify cefazolin concentrations in intervertebral disc (IVD) tissue obtained during single-level discectomy and to evaluate their association with the degree of disc degeneration, according to the Pfirrmann grading system, and the type of endplate changes, according to Modic classification.

## 2. Materials and Methods

This was a single centre prospective observational study conducted at the Hospital of Traumatology and Orthopaedics in Riga, Latvia. After approval by the Hospital Ethics Committee, adult patients scheduled for single level microdiscectomy for symptomatic lumbar disc herniation and receiving 2 g of cefazolin as part of their perioperative antibiotic regimen were consecutively enrolled after providing written informed consent. The enrollment period was from November 2021 to June 2023. Patients were excluded if they had received any antibiotic treatment within the previous 2 weeks, had active infection or known allergy to cefalosporins. All patients underwent an MRI at baseline. We collected demographic and clinical data, including: sex, age, weight, affected intervertebral segment, smoking status, and the presence of diabetes or renal impairment.

### 2.1. Antibacterial Prophylaxis Protocol

In accordance with local hospital guidelines based on international recommendations [4], patients received prophylactic cefazolin (2 g) administered intravenously within 60 min before skin incision. No additional doses were given prior to the collection of serum and disc samples.

### 2.2. Sample Collection

To obtain IVD material for measurement of cefazolin concentration, a rectangular window was created in both the posterior longitudinal ligament and anulus fibrosus during the surgical procedure, and a portion of the nucleus pulposus was excised while attempting to minimize contamination of the tissue by blood. The disc specimen was collected immediately after excision, placed into a sterile container, and the collection time was recorded. The specimen was stored at −80 °C until analysis.

Blood sampling to measure the plasma concentration of cefazolin was performed at two different time points. The baseline sample was taken at the time of peripheral venous catheterization prior to anesthesia. The second blood sample was taken 30 min after the administration of cefazolin. Blood samples were collected in EDTA tubes (5 mL). Within 15 min after sampling, the EDTA tubes were centrifuged at 3500× *g* for 10 min to obtain plasma. Plasma was stored at −80° C until analysis. All three time points, including the baseline blood sample, the second sample, and the time of cefazolin administration, were recorded.

### 2.3. MRI

Two experienced orthopedic surgeons independently graded the severity of IVD and endplate changes using MRI images. Grading of intervertebral disc degeneration was performed on sagittal T2WI images, and the Pfirrmann criteria [11] were used to assess the involved intervertebral discs. The Pfirrmann grading system evaluates degenerated intervertebral discs by MRI for asymmetry in disc structure, distinction of the nucleus and annulus, signal intensity of IVDs, and height of IVDs, and assigns a grade from I to V for disc degeneration. The upper and lower endplate of the involved disc were graded according to Modic classification: type 0, indicating normal endplate; type I, indicating endplate neovascularity that was hyperintense on T2-weighted images and hypointense on T1-weighted images; type II, indicating endplate fatty replacement that was hyperintense on T1-weighted images and isointense or hypointense on T2-weighted images; and type III, indicating endplate bony sclerosis that was hypointense on both T1- and T2-weighted images [12]. Interobserver agreement was evaluated using Cohen’s kappa coefficient, and any discrepancies were resolved through joint review.

### 2.4. Follow Up

To monitor for postoperative discitis, patients underwent follow-up appointments at 2 and 4 months.

### 2.5. Sample Analysis

IVD samples were thawed, cut, and 50–400 mg portions were weighed into 2 mL Eppendorf tubes. After adding 1 mL grinding buffer (25 mM NaH_2_PO_4_, pH 3), samples were homogenized with a glass rod, sonicated for 10 min in an ultrasonic bath filled with ice, and centrifuged (9000× *g*, 10 min). The supernatant was filtered through PVDF Ultrafree-MC filters (Merck Milipore Ltd., Carrigtwohill, Ireland) and transferred to high-performance liquid chromatography (HPLC) vials with inserts. External calibration was used.

Plasma samples were thawed at room temperature. Proteins were precipitated by adding 50 µL water and 600 µL cold acetonitrile to 200 µL plasma, followed by centrifugation (9000× *g*, 10 min). The supernatant was then mixed with 600 µL trichloromethane, vortexed, and centrifuged (14,000× *g*, 2 min). The upper aqueous layer was transferred to HPLC vials with inserts. Matrix-matched calibration was performed using cefazolin-spiked blank plasma.

Cefazolin concentrations were determined by HPLC using a ThermoScientific Ultimate 3000 system (Thermo-Fisher Scientific, Les Ulis, France). Separation was performed using an Ascentis Express C18 column (2.7 µm, 100 mm × 4.6 mm; Merck, Darmstadt, Germany) with a guard column. The column was thermostated at 40 °C, and a 2 µL sample was injected into a mobile phase of acetonitrile (A) and 25 mM phosphate buffer pH 3 at 1.4 mL min^−1^. Gradient conditions were such that A was increased linearly from 12% to 70% over a period of 5.5 min and was kept at 70% for another 3 min. Total analysis run time was 11 min, and the UV detector was set at 272 nm.

The lower limit of quantitation for cefazolin concentration was 1.62 µg mL^−1^ (linear range 1.62–245 µg mL^−1^), with interday and intraday coefficients of variation < 5%.

### 2.6. Primary Outcomes

The primary outcome of the study was the IVD cefazolin concentration (µg g^−1^) in relation to degeneration severity and endplate changes, as quantified by MRI. To evaluate the efficacy of prophylactic cefazolin, the proportion of IVD samples with cefazolin concentrations exceeding target thresholds of 4, 8, and 16 µg/g was calculated. These thresholds were based on the MIC_90_ values (minimum inhibitory concentration at which 90% of tested strains are inhibited) for *Staphylococcus aureus*, *Staphylococcus epidermidis*, and *Enterococcus* spp., the most common pathogens associated with SSIs.

### 2.7. Statistical Analysis

Data were analyzed using statistical software R (version 4.0.5). Baseline characteristics are presented for the entire cohort. Continuous variables are presented as mean (standard deviation), categorical data as number and percentage.

Categorical variables were compared using the Fisher’s exact test. A heteroscedastic ANOVA model was used to compare cefazolin concentrations between different disc degeneration grades and endplate change types, with Pfirrmann grade V and Modic type 0 as reference groups for comparisons. *p*-values and 95% confidence intervals were adjusted for multiple testing using the max-t test method, as implemented in the multcomp package.

Effects of patient age, weight, and time from cefazolin administration to IVD sampling on plasma and IVD cefazolin concentrations were analyzed using general linear regression modeling.

A two-sided *p* value < 0.05 indicated statistical significance.

For our study to have 80% power to detect a 4 µg g^−1^ difference in mean cefazolin concentration between three grades of disc degeneration with a significance level of 0.05, 12 participants per degeneration subgroup were required. To ensure adequate power after accounting for potential patient drop out and subgroup imbalance, a total of 96 patients were enrolled.

## 3. Results

Between November 2021 and June 2023, 106 patients were enrolled in the study. Of the 106 collected samples, 4 were excluded from analysis due to analytical errors (*n* = 3) and improper transport conditions (*n* = 1), resulting in 102 valid IVD sample measurements. In addition, MRI images were unavailable for 6 patients, who were excluded from degeneration grading.

The characteristics of the study population are summarized in Table 1. Most participants (62.5%) had moderate intervertebral disc degeneration (Pfirrmann grade IV), and none had discs classified as normal (Pfirrmann grade ≤ II). Modic changes were present in 49.6% of patients, with type I changes observed in 4.7% and type II in 44.9%. Interobserver agreement between the two MRI evaluators was 0.86 for Pfirrmann grading and 0.82 for Modic changes. Two patients developed surgical site infections; their IVD cefazolin concentrations were 4.6 (Pfirrmann III, MODIC II) and 13.3 µg g^−1^ (Pfirrmann III, No MODIC).

Cefazolin was undetectable in serum prior to antibiotic administration. Thirty minutes after administration, the plasma concentration of cefazolin was 110.9 ± 31.2 mg L^−1^. The mean time to collection of IVD samples was 73 ± 42 min after antibiotic administration. The mean concentration of cefazolin in IVDs was 12.1 ± 7.2 µg g^−1^, with a range of 2.0–43.3 µg g^−1^.

Plasma cefazolin concentrations correlated with weight (r^2^ = 0.14, *p* < 0.001). No correlations were found between the IVD concentration of cefazolin and patient age (r^2^ = 0.01), weight (r^2^ = 0.01), or time from cefazolin administration to IVD sampling (r^2^ = 0.03).

Table 2 demonstrates the mean serum and IVD concentrations of cefazolin in samples obtained from patients with different disc degeneration grades. The mean IVD cefazolin concentrations were 11.3 ± 8.7 µg g^−1^ for Pfirrmann III, 11.7 ± 7.1 µg g^−1^ for Pfirrmann IV, and 14.9 ± 6.0 µg g^−1^ for Pfirrmann V (*p* = 0.29). There were no differences in patient weight (r^2^ = 0.01) or time from cefazolin administration to IVD sampling (r^2^ = 0.02) between different disc degeneration grades.

The effect of Modic change type on IVD concentrations of cefazolin is shown in Table 3. The IVD concentrations were significantly higher in cases with Modic type II changes compared to those without or type I changes (14.6 ± 9.2 µg g^−1^ vs. 10.2 ± 4.5 µg g^−1^ and 9.2 ± 4.1 µg g^−1^; *p* = 0.01). There were no differences in patient weight (r^2^ = 0.02) or time from cefazolin administration to IVD sampling (r^2^ = 0.01) between different endplate change types.

Ninety-four percent of the participants exhibited cefazolin concentrations in IVD of >4 µg g^−1^. However, this percentage decreased for concentrations >8 and >16 µg g^−1^. The distribution of samples above and below thresholds of 4, 8 and 16 µg cefazolin/g tissue in each degeneration grade and endplate type is shown in Figure 1. For patients with Modic type 0 or type I changes, only 10% and 25% had concentrations > 16 µg g^−1^, compared to 35.4% for type II (*p* = 0.008). Similarly, only 16.7% and 20.3% of Pfirrmann grade III and IV discs had concentrations > 16 µg g^−1^, compared to 34.6% for Pfirrmann grade V discs (*p* = 0.26).

## 4. Discussion

Our results indicate that mean IVD cefazolin concentrations in patients undergoing single level discectomy were higher in those with more advanced disc degeneration (Pfirrmann grade V) and Modic type II endplate changes. Using 4 µg g^−1^ as a target, six percent of patients did not achieve cefazolin concentrations above the proposed thresholds for effective antimicrobial activity.

Previous studies have investigated the penetration of cefazolin into cervical and lumbar IVDs and reported varying penetration rates. Cefazolin concentrations in these studies have been expressed in both μg g^−1^ and mg L^−1^. Given that the nucleus pulposus comprises approximately 83% water by tissue weight and has a density is close to 1.0, the results can be compared directly across studies [14]. In the current study, the mean IVD cefazolin concentration was 12.1 ± 7.2 μg g^−1^ with a range of 2.7 –43.1 μg g^−1^. These values fall between lower concentrations of 0.2 –0.8 mg L^−1^ [8] and 2.33 ± 0.45 μg g^−1^ [15], and higher concentrations of 59.91 ± 25.79 μg mL^−1^ [16] reported in prior studies. While one study reported that only 50% of tissue samples achieved the minimum inhibitory concentration for *Staphylococcus aureus* following administration of 1 g cefazolin prophylaxis [8], our results show that 94% of patients reached this threshold using the 2 g dosage. Although Walters et al. have suggested that variability of IVD cefazolin concentrations could be explained by factors such as disc degeneration and disc size, previous studies have not addressed this issue [8]. Our study extends previous observations by supporting an association between IVD and endplate degeneration and better IVD cefazolin penetration.

The cartilaginous endplate, which interfaces the disc and bone, influences diffusion of nutrients and therapeutic agents into IVD [17,18]. Degeneration of this interface, seen as Modic changes on MRI, could impact delivery of antibiotics to affected discs. In this study, the observed variation in cefazolin IVD concentrations across different types of Modic changes may be attributed to altered endplate morphology. Modic Type II changes involve chronic inflammation, fibrosis, and angiogenesis. The presence of inflammatory markers and angiogenic factors, such as vascular endothelial growth factor (VEGF) and stromal cell-derived factor-1α (SDF-1α), suggests that neovascularization occurs in response to inflammation and tissue damage [19]. This neovascularization, together with associated inflammation, may enhance local blood supply and facilitate improved antibiotic delivery, thereby increasing cefazolin penetration into the affected discs.

Some studies also indicate that charge plays a significant role in the kinetics of antibiotic penetration into the IVD [20]. As IVD degeneration progresses, reduction in proteoglycan content leads to a decrease in the disc’s negative charge. The less negative charge environment in IVDs with higher Pfirrmann grades may improve the penetration of negatively charged antibiotics, such as cefazolin, into degenerated disc tissue. However, our findings only partially support this hypothesis, though higher cefazolin concentrations were observed in discs with advanced degeneration, the difference was not statistically significant. An animal study by Walters et al. found no difference in cefazolin penetration between normal and artificially degenerated sheep discs [21]. Therefore, the effect of IVD degeneration on cefazolin penetration remains inconclusive.

The most common cause of postoperative IVD infections is *Staphylococcus aureus* (60%), and less commonly *Escherichia coli*. According to the European Committee on Antimicrobial Susceptibility Testing (EUCAST), the minimum inhibitory concentration of cefazolin required to inhibit growth susceptible strains of these bacteria is 1 μg mL^−1^ (European Committee on Antimicrobial Susceptibility Testing. Data from the EUCAST MIC distribution website, last accessed 4 January 2025). Four to five times higher values (4 μg mL^−1^) are required for clinical efficacy [22]. Our study found no significant difference in attainment of concentration needed to cover standard microorganisms across all degrees of IVD degeneration and Modic types. Yet, Modic type II changes significantly increased attainment of higher cefazolin concentrations (>16 μg mL^−1^) needed to cover Gram-negative organisms. However, the relatively small sample size of patients with Modic type I changes compared to other Modic groups limits our ability to draw definitive conclusions about antibiotic penetration in this population.

As the primary objective of perioperative antibacterial prophylaxis is to prevent iatrogenic spondylodiscitis, IVD cefazolin concentrations can only serve as a surrogate outcome. A direct correlation between local antibiotic concentrations and surgical site infection rates has not yet been demonstrated. Establishing a direct causal relationship between IVD cefazolin levels and the incidence of postoperative spondylodiscitis would require a large, multicenter prospective study involving approximately 3000 patients to link tissue concentrations with clinical outcomes and determine whether personalized antibiotic dosing strategies could improve their prophylactic efficacy in spine surgery.

We acknowledge several limitations to our study. One limitation of our methodology is analysis of total as opposed to free cefazolin concentrations. However, our method provided a comprehensive assessment of cefazolin penetration into IVDs and total cefazolin concentrations are clinically relevant as they correlate with microbiological outcomes in patients [23]. Another limitation is measurement of tissue cefazolin concentrations via homogenized samples which may underestimate interstitial fluid concentration as intracellular fluid, lacking cefazolin, is included in the homogenate. Additionally, there was heterogeneity in IVD sampling time; however, time from cefazolin administration to IVD sampling did not correlate with IVD cefazolin concentrations. Pharmacokinetic confounders, such as age, BMI, and renal function, were comparable between groups and are therefore unlikely to have biased the results. The distribution of degeneration grades was uneven (22, 60, and 14 patients for Pfirrmann grades III, IV, and V, respectively). Although the study had adequate power for the overall ANOVA (75–95%), some pairwise comparisons involving Pfirrmann V subgroup may have been underpowered. Therefore, the absence of statistically significant differences in cefazolin concentrations across Pfirrmann grades could reflect limited sample size rather than a true lack of effect. Our study also focused on degenerated discs, which is a clinically relevant population, and we acknowledge that investigating normal discs would have provided valuable additional information.

Cefazolin continues to be widely used in surgical prophylaxis in spinal surgery, but its limited penetration into IVD tissue due to avascularity and matrix barriers necessitates careful dosing, timing, and consideration of patient-specific factors. While emerging evidence suggests that the use of individualized regimens could help to overcome these challenges, the clinical relevance of the differential cefazolin penetration observed in patients with different types of Modic changes is unclear. Further clinical comparative effectiveness trials and integration of advanced pharmacokinetic modelling are needed to clarify the impact of endplate pathology on antibiotic penetration and to evaluate whether individualized dosing strategies could improve the efficacy of perioperative prophylaxis in spine surgery.

## 5. Conclusions

While ninety-four percent of intervertebral disc samples achieved cefazolin concentrations exceeding the minimum inhibitory concentration for *Staphylococcus aureus* (>4 µg g^−1^), this study demonstrates substantial variability in intradiscal cefazolin levels following a fixed preoperative dosing regimen, with higher concentrations observed in discs exhibiting Modic type II changes.

## Figures and Tables

**Figure 1 medicina-61-01999-f001:**
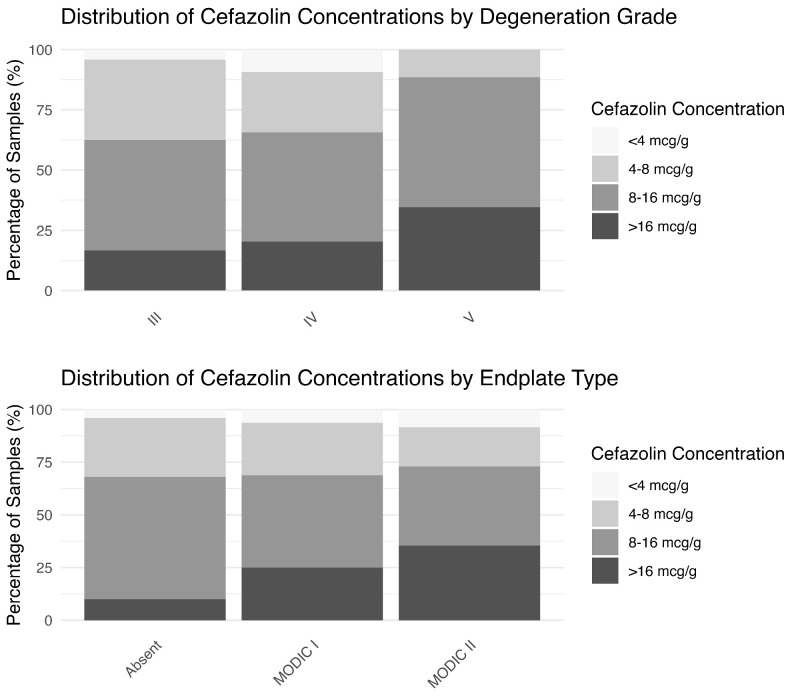
Percentage of patients whose cefazolin concentrations were above different concentration targets.

**Table 1 medicina-61-01999-t001:** Summary of participant characteristics. BMI, body mass index BMI, eGFR, estimated glomerular filtration rate.

Variable	Overall (*n*= 96)
Age (years)	45.9 (12.6) ^1^
Sex (male, n (%))	54 (56.3%)
Height (m)	176 (9.40) ^1^
Weight (kg)	84.2 (17.8) ^1^
BMI (kg/m^2^)	27.0 (4.77) ^1^
eGFR (mL/kg/1.73 m^2^)	103 (27.1) ^1^
Smoker	41 (38.7%)
Diabetes	6 (6.3%)
Intervertebral disc level	
L2–L3	5 (5.2%)
L3–L4	8 (8.3%)
L4–L5	42 (43.8%)
L5–S1	41 (38.7%)

^1^ Data are presented as mean ± SD.

**Table 2 medicina-61-01999-t002:** Plasma and Intervertebral Disc Concentrations of Cefazolin by Degree of Disc Degeneration.

Outcome	Degree of Disc Degeneration			
	Pfirrmann III (*n* = 22)	Pfirrmann IV (*n* = 60)	Pfirrmann V (*n* = 14)	*p* ^1^
Serum (micrograms/mL)	104.4 (35.6)	115.0 (30.1)	101.2 (24.7)	0.24
Disc concentration (micrograms/g)	11.3 (8.7)	11.6 (7.1)	14.9 (6.0)	0.29
Disc-to-plasma concentration ratio	0.11 (0.08)	0.11 (0.07)	0.15 (0.12)	0.37
Patients with disc concentrations less than 4 micrograms/g	1/22	5/60	0/14	0.43

Data are presented as mean (standard deviation) or N. ^1^ Compared with Pfirrmann V group.

**Table 3 medicina-61-01999-t003:** Plasma and Intervertebral Disc Concentrations of Cefazolin by Degree of Endplate Changes.

Outcome	Degree of Endplate Changes			
	Absent (*n* = 45)	MODIC I (*n* = 8)	MODIC II (*n* = 43)	*p* ^1^
Serum (micrograms/mL)	111.0 (39.4)	107.2 (24)	111.5 (20.9)	0.94
Disc concentration (micrograms/g)	10.1 (4.5)	9.2 (4.1)	14.6 (9.2)	0.01
Disc-to-plasma concentration ratio	0.11 (0.08)	0.11 (0.07)	0.15 (0.12)	0.37
Patients with disc concentrations less than 4 micrograms/g	2/45	1/8	3/43	0.54

Data are mean ± standard deviation or N. ^1^ Compared with No MODIC group.

## Data Availability

The data supporting the conclusions of this study will be made available by the authors upon request.

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
