# Peer review of "Impact of Intervertebral Disc Degeneration and Endplate Changes on Cefazolin Penetration into the Intervertebral Disc"

_medicina, 2025, doi:10.3390/medicina61111999_

Round 1
Reviewer 1 Report
Comments and Suggestions for Authors
This manuscript addresses an important and clinically relevant topic regarding cefazolin penetration into intervertebral discs in relation to disc degeneration and Modic changes. The study has scientific merit, an adequate sample size, and well-presented results. However, there are significant issues regarding the clarity of the study rationale, methodological transparency, and depth of interpretation. Major revisions are required before the manuscript can be considered for publication.
However, several important comments and improvements need to be addressed before the manuscript can be accepted for publication. The following numbered points outline the key issues that require revision.
- Introduction
The introduction is too brief and does not provide an adequate framework for the problem. It should better explain:
- The clinical relevance of postoperative discitis and antibiotic prophylaxis in spine surgery.
- The pharmacological challenge of drug penetration into the avascular intervertebral disc.
- Why disc degeneration and Modic changes could theoretically influence antibiotic transport.
- The research gap and a clearer hypothesis.
A more comprehensive introduction will improve scientific context and reader orientation.
- Study Design and Patient Selection
The study design is not explicitly stated and should be clarified as a prospective observational study. Inclusion and exclusion criteria are incomplete. Aside from prior antibiotic exposure, there is no mention of comorbidities such as diabetes, renal impairment, obesity, or smoking, all of which may alter antibiotic pharmacokinetics. These criteria must be defined to ensure methodological transparency and reduce selection bias.
- Antibiotic Prophylaxis Protocol
The antibiotic regimen (2 g cefazolin) is consistent with common prophylactic standards; however, the authors do not justify this dosage or explain whether the timing relative to the incision was standardized. It is unclear if intraoperative redosing was performed and whether dosing was weight-adjusted. These missing details may influence intradiscal cefazolin levels and should be clarified.
- Data Collection Standards
Important aspects of data quality and standardization are missing. There is no explanation of how timing between cefazolin administration and disc sampling was controlled, yet this variable has a strong pharmacokinetic influence on tissue levels. The MRI grading was performed by two observers, but it is not stated whether the observers were blinded to cefazolin levels. Interobserver reliability was also not reported. The authors should clarify whether blinding was used to reduce assessment bias and include interobserver agreement data.
- Statistical Analysis
The statistical methods are generally appropriate, but the manuscript lacks a justification for the sample size. The rationale for using threshold cefazolin values (4, 8, 16 µg/g) should be explained in the Methods section, rather than being introduced for the first time in the Results.
- Discussion
The discussion does not sufficiently explain why Modic type II changes may increase cefazolin penetration. Potential mechanisms such as vascular fatty infiltration, microstructural endplate disruption, and altered tissue permeability should be considered. Clinical implications should also be discussed—does this variability suggest a need for personalized antibiotic dosing in spine surgery? The limitations section should address heterogeneity in sampling time and lack of control for pharmacokinetic confounders.
- Conclusions
The conclusions should be more cautious. The results demonstrate associations but cannot infer causality. The authors should avoid suggesting clinical recommendations without adequate evidence and instead emphasize directions for future research.
Overall, this is a well-conceived and clinically relevant study that addresses an important question in spinal surgery antibiotic prophylaxis. With clarification of the methodology, expansion of the introduction, and a more comprehensive discussion of the findings, the manuscript has clear potential for publication after major revisions.
Author Response
Response to Reviewer 1
- Introduction
Comment: The introduction is too brief and does not provide an adequate framework for the problem. It should better explain:
- The clinical relevance of postoperative discitis and antibiotic prophylaxis in spine surgery.
- The pharmacological challenge of drug penetration into the avascular intervertebral disc.
- Why disc degeneration and Modic changes could theoretically influence antibiotic transport.
- The research gap and a clearer hypothesis.
A more comprehensive introduction will improve scientific context and reader orientation.
Response: We have rewritten the introduction section.
- Study Design and Patient Selection
Comment: The study design is not explicitly stated and should be clarified as a prospective observational study.
Response: We have clarified the study design in the manuscript (line 104).
Comment: Inclusion and exclusion criteria are incomplete. Aside from prior antibiotic exposure, there is no mention of comorbidities such as diabetes, renal impairment, obesity, or smoking, all of which may alter antibiotic pharmacokinetics. These criteria must be defined to ensure methodological transparency and reduce selection bias.
Response: We have added detailed exclusion criteria to the manuscript. We agree with the reviewer's point that excluding patients with co-morbidities or obesity can improve internal validity by reducing pharmacokinetic variability. Yet, such exclusions can compromise external applicability by omitting patient groups commonly encountered in clinical practice. Therefore, we included all patients presenting for single level microdiscectomy. The incidence of diabetes, renal impairment and obesity in this relatively young patient population was very low. We have added information regarding co-morbidities to Table 1 (line 107-109).
- Antibiotic Prophylaxis Protocol
Comment: The antibiotic regimen (2 g cefazolin) is consistent with common prophylactic standards; however, the authors do not justify this dosage or explain whether the timing relative to the incision was standardized. It is unclear if intraoperative redosing was performed and whether dosing was weight-adjusted. These missing details may influence intradiscal cefazolin levels and should be clarified.
Response: We have added a justification for the choice of cefazolin to the manuscript and clarified the timing of administration (within 60 minutes before incision). Information regarding redosing has also been included: given the duration of the operations, no additional doses were administered before serum or disc sampling. According to local guidelines, the cefazolin dose should be increased in patients weighing more than 120 kg; however, as none of the study participants exceeded this weight, no dose adjustments were required (lines 112-116).
- Data Collection Standards
Comment: Important aspects of data quality and standardization are missing. There is no explanation of how timing between cefazolin administration and disc sampling was controlled, yet this variable has a strong pharmacokinetic influence on tissue levels.
Response: We agree that the timing between cefazolin administration and disc sampling is an important factor influencing tissue concentration. In the clinical setting, however, this interval cannot be precisely controlled due to variations in surgical duration and workflow. Nevertheless, the exact timing between cefazolin administration and disc sampling was recorded for each patient and included in the subsequent data analysis to account for its potential influence on tissue levels.
Comment: The MRI grading was performed by two observers, but it is not stated whether the observers were blinded to cefazolin levels. Interobserver reliability was also not reported. The authors should clarify whether blinding was used to reduce assessment bias and include interobserver agreement data.
Response: The MRI grading was performed independently by two orthopaedic surgeons who were blinded to the cefazolin concentration data. Interobserver agreement has now been reported in the revised manuscript (lines 144-146).
- Statistical Analysis
Comment: The statistical methods are generally appropriate, but the manuscript lacks a justification for the sample size. The rationale for using threshold cefazolin values (4, 8, 16 µg/g) should be explained in the Methods section, rather than being introduced for the first time in the Results.
Response: Threshold cefazolin concentrations of 4, 8, and 16 µg/g were selected based on the antibiotic resistance breakpoints for Staphylococcus aureus, Staphylococcus epidermidis and Enterococcus spp., reflecting clinically relevant levels associated with susceptibility to cefazolin. We have added appropriate explanations to the Methods section (lines 172-179).
- Discussion
Comment: The discussion does not sufficiently explain why Modic type II changes may increase cefazolin penetration. Potential mechanisms such as vascular fatty infiltration, microstructural endplate disruption, and altered tissue permeability should be considered.
Response: We have expanded the discussion to address potential mechanisms by which Modic Type II changes may increase cefazolin penetration. Modic Type II changes involve chronic inflammation, fibrosis, and angiogenesis. The presence of inflammatory markers and angiogenic factors, such as vascular endothelial growth factor (VEGF) and stromal cell-derived factor-1α (SDF-1α), suggests that neovascularization occurs in response to inflammation and tissue damage. This neovascularization, together with associated inflammation, may enhance local blood supply and facilitate improved antibiotic delivery, thereby increasing cefazolin penetration into the affected discs.
Comment: Clinical implications should also be discussed—does this variability suggest a need for personalized antibiotic dosing in spine surgery?
Response: While the variability in cefazolin concentrations among patients may suggest potential for personalized antibiotic dosing, a direct correlation between tissue concentrations and surgical site infection rates has not yet been established. We have revised the Discussion to clarify that (lines 306-313).
Comment: The limitations section should address heterogeneity in sampling time and lack of control for pharmacokinetic confounders.
Response: We have addressed this point in the revised Discussion section. Although heterogeneity in sampling time was present, no significant association was found between sampling time and IVD cefazolin concentration. Furthermore, pharmacokinetic confounders such as age, BMI, and renal function did not differ significantly across the different degrees of disc degeneration or types of endplate changes (lines 321-324).
- Conclusions
Comment: The conclusions should be more cautious. The results demonstrate associations but cannot infer causality. The authors should avoid suggesting clinical recommendations without adequate evidence and instead emphasize directions for future research.
Response: The conclusion has been revised to adopt a more cautious interpretation of our findings (line 345-348).
Reviewer 2 Report
Comments and Suggestions for Authors
The authors examine the impact of perioperative intravenous antibiotic in degenerated discs and vertebral endplates. The idea is good and the design of the study simple but adequate.
I have some points to discuss and/or change:
1) in your methods you should be more detailed in the inclusion criteria. In which hospital, how many patients and of course exclusion criteria.
2) it is mostly impossible to take nucleus pulposus without any blood on it at all. So I understand what you tried but please change the expression in "we tried" or something similar.
3) What made the 4 samples out of 106 invalid?
4) line 167 was supposed to be a chapter title or what? Please correct accordingly.
5) BMI of the patients is an important factor and should be taken under serious consideration as covariate. Higher BMI could mean more antibiotics needed.
6) The fact that more than half of your cohort had a Pfirrmann IV disc degeneration may have influenced your results. Please discuss or add in limitations.
7) In your cohort you had two cases of postoperative infection. Is it possible to find the grade of disc degeneration, Modic and antibiotic concentration in the disc for these cases? It would be interesting.
8) In your limitations should be mentioned the relatively small cohort and the inhomogeneous degeneration grade. You also miss a healthy group but of course that wouldn't be possible anyway due to ethical concerns.
9) I miss your theory on how your results are explained. Why do we have more antibiotic penetration to the degenerated discs? Due to lower saturation of the disc or due to increased perfusion because of the bone inflammation? Thats the most important in your work, the interpretation of the results.
10) You gave 2g Cefazolin. Why not 1.5g Cefuroxim? Is it the standard at your center? Both are efficient, no question. I am just curious.
11) So what is your suggestion after having these results? Adapt the antibiotic dosis to the degree of degeneration or something else?
12) tell us about your institution, volume of patients and where were the surgeries done.
Author Response
Response to Reviewer 2
Comment 1: In your methods you should be more detailed in the inclusion criteria. In which hospital, how many patients and of course exclusion criteria.
Response: We have added more detailed exclusion and inclusion criteria to the manuscript. Excluding patients with co-morbidities or obesity can improve internal validity by reducing pharmacokinetic variability. Yet, such exclusions can compromise external applicability by omitting patient groups commonly encountered in clinical practice. Therefore, we included all patients presenting for single level microdiscectomy. The incidence of diabetes, renal impairment and obesity in this relatively young patient population was very low. We have added information regarding co-morbidities to Table 1.
Comment 2: It is mostly impossible to take nucleus pulposus without any blood on it at all. So I understand what you tried but please change the expression in "we tried" or something similar.
Response: We agree that complete avoidance of blood contamination is technically difficult. The sentence has been revised to more accurately reflect the surgical procedure (lines 118-122).
Comment 3: What made the 4 samples out of 106 invalid?
Response: Four samples were excluded due to analytical and handling issues. Three samples were excluded because of analytical errors during laboratory processing, and one was excluded due to improper transport conditions that could have affected the sample. (lines 195-199)
Comment 4: line 167 was supposed to be a chapter title or what? Please correct accordingly.
Response: The issue has been corrected.
Comment 5: BMI of the patients is an important factor and should be taken under serious consideration as covariate. Higher BMI could mean more antibiotics needed.
Response: BMI was evaluated as a potential covariate in our analysis. No significant correlation was found between BMI and IVD cefazolin concentrations (r² = 0.02, p = 0.18). Moreover, BMI did not differ significantly between patients with different degrees of disc degeneration or Modic endplate changes. This suggests that within the studied population, BMI had no measurable effect on cefazolin penetration into intervertebral disc tissue within the timeframe of sampling.
Comment 6: The fact that more than half of your cohort had a Pfirrmann IV disc degeneration may have influenced your results. Please discuss or add in limitations.
Response: We agree that the skewed distribution of Pfirrmann grades could influence results. Using our power calculation assumptions (within-group SD = 6 µg/mL and adjacent group difference = 4 µg/mL), the observed sample (Pfirrmann III = 22, IV = 60, V = 14; N = 96) yields 94.7% power for ANOVA (α = 0.05). Power for specific pairwise contrasts is lower: III vs IV 75%, IV vs V 65%, and III vs V 97%. Thus, while the overall test is well powered, comparisons that involve Pfirrmann V group should be interpreted with caution. This has been added to Limitations.
Comment 7: In your cohort you had two cases of postoperative infection. Is it possible to find the grade of disc degeneration, Modic and antibiotic concentration in the disc for these cases? It would be interesting.
Response: The information has been added to line 212.
Comment 8: In your limitations should be mentioned the relatively small cohort and the inhomogeneous degeneration grade. You also miss a healthy group but of course that wouldn't be possible anyway due to ethical concerns.
Response: While we acknowledge that the cohort size could be considered relatively small and that there was some heterogeneity in degeneration grades, our study included 96 patients—substantially more than previously published studies on intradisc cefazolin concentrations, such as Walters et al. (n=30) and Capoor et al. (n=25). Therefore, to our knowledge, this represents the largest clinical cohort to date investigating cefazolin penetration into intervertebral disc.
Regarding the absence of a healthy control group, we fully agree with the reviewer. Sampling non-degenerated intervertebral discs would require surgical intervention in asymptomatic individuals, which would not be ethically justifiable. Our analysis was therefore restricted to surgically removed, clinically degenerated discs.
Comment 9: I miss your theory on how your results are explained. Why do we have more antibiotic penetration to the degenerated discs? Due to lower saturation of the disc or due to increased perfusion because of the bone inflammation? Thats the most important in your work, the interpretation of the results.
Response: We have expanded the discussion to address potential mechanisms by which Modic Type II changes may increase cefazolin penetration. Modic Type II changes involve chronic inflammation, fibrosis, and angiogenesis. The presence of inflammatory markers and angiogenic factors, such as vascular endothelial growth factor (VEGF) and stromal cell-derived factor-1α (SDF-1α), suggests that neovascularization occurs in response to inflammation and tissue damage. This neovascularization, together with associated inflammation, may enhance local blood supply and facilitate improved antibiotic delivery, thereby increasing cefazolin penetration into the affected discs.
Comment 10: You gave 2g Cefazolin. Why not 1.5g Cefuroxim? Is it the standard at your center? Both are efficient, no question. I am just curious.
Response: National and local hospital clinical guidelines recommend cefazolin for surgical prophylaxis in orthopaedic/spine surgery because of its proven efficacy, safety profile, and cost-effectiveness. Cefazolin is also associated with a lower risk of promoting antibiotic resistance compared with cefuroxime.
Comment 11: So what is your suggestion after having these results? Adapt the antibiotic dosis to the degree of degeneration or something else?
Response: Our results demonstrate association of higher cefazolin concentrations in discs with Modic type II changes but cannot infer causality. The clinical significance of this association remains uncertain. Further studies are needed to to evaluate whether individualized dosing strategies could improve the efficacy of perioperative prophylaxis in spine surgery.
Comment 12: Tell us about your institution, volume of patients and where were the surgeries done.
Response: All operations were performed at the Hospital of Traumatology and Orthopaedics in Riga, the largest spine surgery centre in Latvia, which performs approximately 350 spine surgeries annually, including discectomies, decompressions, and spinal fusions.
Round 2
Reviewer 2 Report
Comments and Suggestions for Authors
Thanks for the changes. I am satisfied with the result.